# A ceratopsid-dominated tracksite from the Dinosaur Park Formation (Campanian) at Dinosaur Provincial Park, Alberta, Canada

**Phil R. Bell**[1⦿], **Brian J. Pickles**[2⦿*], **Sarah C. Ashby**[2], **Issy E. Walker**[2], **Sally Hurst**[3], **Michael Rampe**[4], **Paul Durkin**[5], **Caleb M. Brown**[5,6,7]

**1** Palaeoscience Research Centre, School of Environmental and Rural Science, University of New England, Armidale, New South Wales, Australia, **2** School of Biological Sciences, University of Reading, Whiteknights, Reading, United Kingdom, **3** School of Natural Sciences, Macquarie University, Macquarie Park, New South Wales, Australia, **4** Faculty of Arts, Macquarie University, Macquarie Park, New South Wales, Australia, **5** Department of Earth Sciences, University of Manitoba, Winnipeg, Manitoba, Canada, **6** Royal Tyrrell Museum of Palaeontology, Drumheller, Alberta, Canada, **7** Department of Biological Sciences, University of Alberta, Edmonton, Alberta, Canada

⦿ These authors contributed equally to this work.

* b.j.pickles@reading.ac.uk

## Abstract

The badlands of Dinosaur Provincial Park (Alberta, Canada) are renowned for the exceptional abundance and diversity of Campanian-aged vertebrate body fossils, especially dinosaurs. Due to the steep exposures and rapid erosion, dinosaur tracks and trackways are considered extremely rare but have been recorded from a small number of concretionary casts, which pertain to hadrosaurids and a single tyrannosaurid. Here, we document the first multitaxic dinosaur footprint assemblage from the Dinosaur Park Formation based on a new locality that contains multiple individual ceratopsids, two tyrannosaurids, a possible ankylosaurian, and a small theropod-like taxon. Ceratopsid tracks are globally rare but dominate the new tracksite, suggesting gregarious behaviour, which is also supported by their regular spacing and parallel arrangement. The possible ankylosaurian track is identified (in part) on account of having three distinct pedal digits, consistent with the pedal anatomy of several Dinosaur Park ankylosaurids (*Euoplocephalus, Dyoplosaurus*) and the newly erected ichnotaxon *Ruopodosaurus clava* but differentiating it from other ankylosaurian tracks (*Tetrapodosaurus* isp.). Importantly, the new tracks are the first natural moulds (concave epirelief) found in Dinosaur Provincial Park, which, due to the unique geomorphology of the area, can only be recognised in outcrops where there are prominent sediment displacement rims. The new search image outlined here has already resulted in several subsequent trackway discoveries, and has the potential to transform ichnological studies in the Dinosaur Park Formation and related formations where badlands prevail.

**Data availability statement:** All data necessary to replicate the analyses are presented in the manuscript and its Supporting Information files. The Skyline Tracksite is located within the Dinosaur Provincial Park Palaeontological Preserve. Due to the sensitivity of the site its coordinates cannot be publicly shared but were provided to reviewers. Interested researchers may apply to gain access to the full site coordinate data, which are on file at the Royal Tyrrell Museum of Palaeontology, Drumheller, Alberta. Site number L2467. Please contact RTMP Collections: https://tyrrellmuseum.com/research/collections.

**Funding:** BJP and PRB received a University of New England Visiting Research Fellowship. There is no Grant Number associated with this award. The funders were not involved in the study design, data collection and analysis, decision to publish, or preparation of the manuscript.

**Competing interests:** The authors have declared that no competing interests exist.

## Introduction

Dinosaur Provincial Park in southern Alberta, Canada, is unquestionably one of the premier localities worldwide for understanding Late Cretaceous terrestrial ecosystems [1]. The Park, as it is known, encompasses rocks of the Belly River Group (BRG) including the Oldman and Dinosaur Park formations, which have yielded hundreds of associated and articulated skeletons and huge numbers of isolated bones and teeth, making it a model system for understanding dinosaur evolution, behaviour, biostratigraphy, and palaeoecology [1]. Despite the remarkable abundance of skeletal elements, dinosaur footprints and trackways are surprisingly rare [2–5]. This rarity is unquestionably linked to the steep exposures and high rates of erosion (2–4 mm/year; [6]) typical of badlands geomorphology [7,8] and the unusual occurrence of concretionary tracks—track casts preserved as siderite concretions that protrude above the fine-grained sediments in which the footprint was originally emplaced [5]. Concretionary tracks, which are so far unique to several Late Cretaceous terrestrial deposits in southern Alberta, differ substantially in appearance from typical natural track moulds and casts, which occur as depressions or embossments, respectively, on bedding surfaces. Although they are probably more common than is typically recognised, the tendency of concretionary tracks to rapidly fall apart when exposed is likely linked to the apparent paucity of tracks in the Dinosaur Park Formation [5].

In 2024, one of us (PRB) discovered a new tracksite, the "Skyline Tracksite" comprising 'typical' natural mould (concave epirelief) tracks, which had heretofore not been identified in the Park. The first track was recognised based on an arc-shaped sediment displacement rim that protruded above the surface; excavation of the displacement rim revealed the first ceratopsid track (C1.1) and the adjacent track of a large theropod (T1.2) (Fig 1A). Further excavation exposed at least 20 partial and complete tracks including multiple trackways of ceratopsid and tyrannosaurid dinosaurs (Figs 1B and 2).

Although several tracks and trackways have been previously reported from the Park, nearly all pertain to the concretionary tracks of hadrosaurids (a single tyrannosaurid track has also been reported [3]), which are the most abundant group of dinosaurs in the Park based on isolated skulls and partial or complete skeletons [9]. This new site represents the first multitaxic tracksite from the Park and includes the first ceratopsid, ankylosaurid, and small theropod tracks. Here, we describe the tracks and discuss the significance of this new site as it relates to the relative abundance of these track makers, possible gregarious behaviour, and track preservation at Dinosaur Provincial Park.

## Materials and methods

Fieldwork was conducted under Research and Collection Permit 24-409 (Alberta Parks) and Permit to Excavate Palaeontological Resources 24-047 (Alberta Culture). All necessary permits were obtained for the described study, which complied with all relevant regulations.

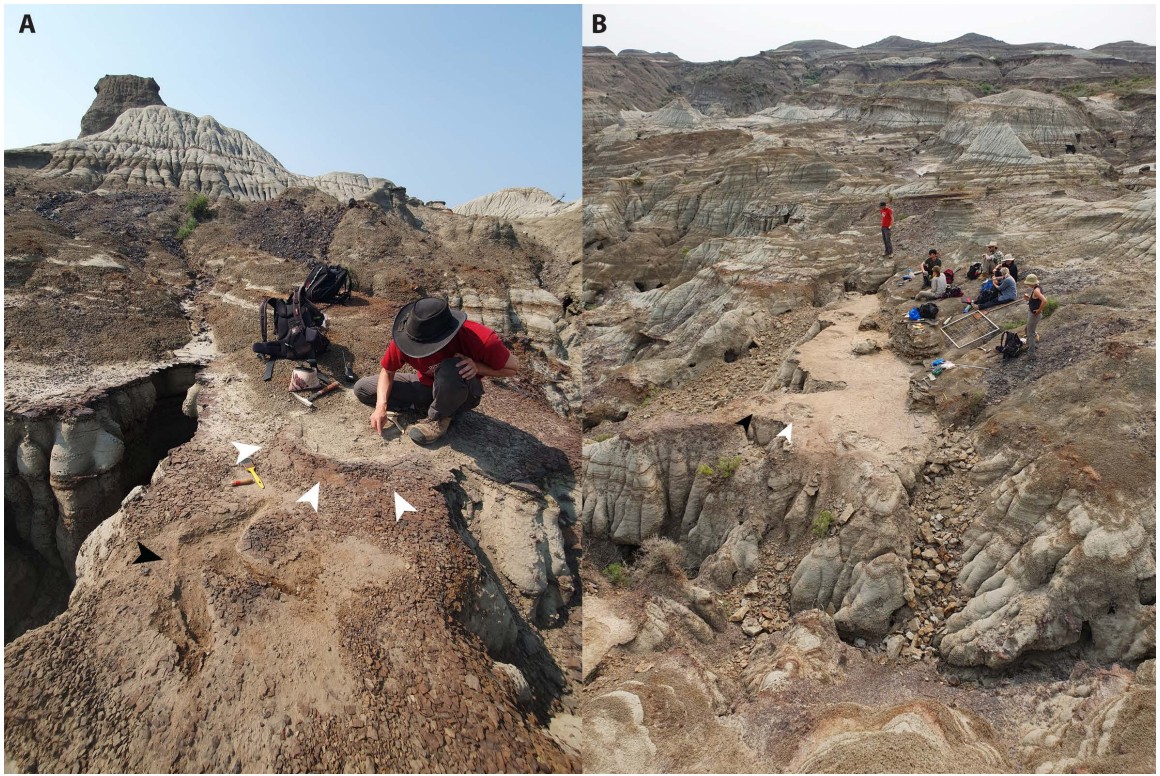

**Fig 1. Views of the Skyline Tracksite (TMP L2467) shortly after discovery (A) and following excavation (B).** A) The sediment displacement rim (white arrows) of what turned out to be the first ceratopsid track (C1.1) was all that was visible when the site was first discovered, view looking west. The inner part of the track was already partly exposed at the time the photo was taken. A tyrannosaurid track (black arrow, T1.2) had also been exposed in the foreground. B) View, looking southeast of the Skyline Tracksite following two days of excavation, note C1.1 and T1.2 indicated by white and black arrows, respectively.

## Excavation methods

The Skyline Tracksite (TMP L2467) is located within the Dinosaur Provincial Park Palaeontological Preserve and is characterized by a resistant iron-cemented horizon overlain by a thick, highly friable mudstone (Fig 1). The site was excavated by hand using picks, awls, trowels, and hammers, to break up the friable mudstone, with the exposed surface cleaned using corn (i.e., whisk) brooms, paintbrushes and dustpans. Rubble was transported to below the track site in an attempt to stabilize the erosional edge of the track horizon and increase the longevity of the site. In some places, the track horizon was capped by a thin (0.5–1 cm) cemented layer, which could be removed in large sheets. In the 2024 field season, ~29 m$^2$ of the main track horizon was exposed, with the unexcavated back wall (15 m long) of the trackway continuing to the south, southeast and southwest. A much smaller excavation (~3 m$^2$) of what is likely the same track horizon took place 24 m to the southeast (125° from baseline), and confirms the continuation of the track site into the unexcavated hill to the south and southeast of the main site. Full site coordinate data are on file at the Royal Tyrrell Museum of Palaeontology, Drumheller, Alberta.

## Mapping methods

Individual tracks were assigned an identifying prefix (A = ankylosaurid; C = ceratopsid; T = tyrannosaurid; U = unknown/unidentified trace) along with a number representing the individual trackmaker (1, 2, etc) and a final sequential number

indicating the position of the track in a trackway (if present). For example, the track number C1.2 identifies the second track in a sequence, or trackway, made by ceratopsid individual #1, whereas T2.1 identifies the first track made by tyrannosaurid individual #2.

The site was mapped using three different methods. Hand-drawn maps were created by first erecting a baseline, which was used to divide the surface into a grid with a portable 1x1 m grid square (quadrat) with 10 cm subdivisions, largely following Rogers [10]. A chalk line was used to mark the baseline and grid boundaries on the trackway surface. Relevant features such as tracks, the erosional edge of the horizon, and extent of the excavation were then drawn onto gridded paper, using the grid square and a plumb bob as a guide. Chalk was used to outline individual tracks to assist in mapping. Hand maps were drawn at two scales (20 cm: 1 m for individual 1x1 m grids, and 1 cm: 1 m for the overview map). The hand-drawn maps were then digitised, scaled, and combined into a single comprehensive map of the locality using Adobe Illustrator (V 23.0.6), which is presented in Fig 2.

The traditional grid square map was augmented using two digital mapping technologies: photogrammetry and differential GPS. Photogrammetric models were made by first photographing the entire track surface with a digital SLR camera, SX10 Mirrorless (Fujifilm: Macquarie Park, NSW, Australia) in an overlapping grid. An initial loop of the site was made, with the camera pointing towards the trackway site. Photos were taken at a constant walking pace, at a similar height and angle on each loop (where possible with varying terrain/elevation), ensuring that each photograph overlapped by at least a third. Additional loops and grid walks were conducted to ensure trackways were photographed from multiple angles and heights, and at a closer range to pick up additional details. Individual tracks were also photographed at a closer range.

A second round of photographs was taken following further excavation at the site a few weeks later using a digital SLR camera, X-S1 (Fujifilm: Bedford, UK) in an overlapping grid. Photos were taken one pace apart, walking the same route four times (forwards and backwards with the camera facing forwards, then forwards and backwards with the camera pointed in the opposite direction), with the camera held at approximately 1.3 m high and pointed downwards at a 45° angle. This process was repeated multiple times.

Overlapping photos (camera settings: SX10 Mirrorless 18 mm lens, X-S1 24 mm lens) were then combined in Agisoft Metashape (version 2.1.3) to digitally reconstruct the tracksite in three-dimensions using standard photogrammetry techniques. These models, both of the entire tracksite, and individual tracks, were uploaded onto the Macquarie University Pedestal 3D gallery (S1 Models: https://mq.pedestal3d.com/r/bePfWqNGay/). To further enhance features of 3D photogrammetric models from Metashape, individual track models were loaded into MeshLab (2023.12) and rendered with a shading filter ('toon.gdp'). In order to allow for the photogrammetry model to be georeferenced and to determine scale and orientation, a series of nine spatial landmarks (large rocks marked with flagging tape) were placed around the tracksite in places that would be captured in the models, but would not obscure trackway features. These marker points were then spatially located using a centimetre-scale differential GPS (Trimble Geo7X, TerraSync). Higher resolution photogrammetric models of individual representative tracks were also captured using a similar method but at a smaller scale.

In addition to the photogrammetric model, relevant features such as baseline posts, baselines, tracks, erosional edge of horizon, and extent of excavation were also traced using a centimetre scale differential GPS (Trimble Geo7X, TerraSync), roughly following the methods of Brown and colleagues [11]. Linear features such as erosional and excavation edges, and partially uncovered tracks were mapped as open lines, while fully exposed tracks were mapped as polygons. Landmark (i.e., vertex) spacing for features ranges from 20–30 cm for erosional and excavation edges, to ~10 cm for tracks. Coordinate data were postprocessed (Trimble Pathfinder) using the Cansel base station in Brooks, Alberta (35 km away), with all points having a postprocessed accuracy of <5 cm.

## Measurement methods

Individual track features were measured in the field. Standard track measurements largely follow Therrien and colleagues [12] and Salisbury and colleagues [13] and included (where possible): compass bearing, track length, track width, free

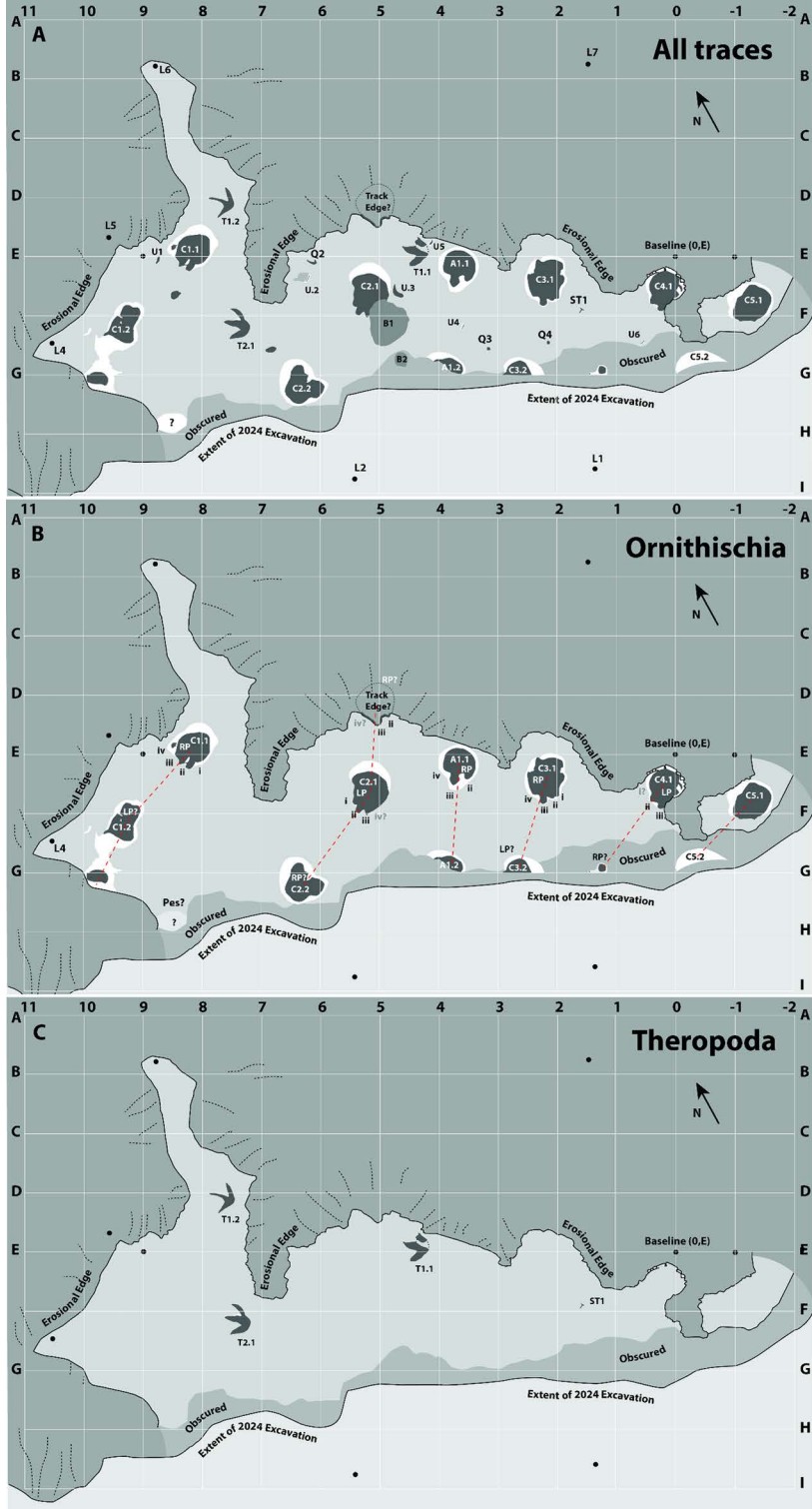

**Fig 2. Schematic line drawing of Skyline Tracksite showing relative position and morphology of the mapped traces as of 2024.** A) All mapped traces. B) Traces likely pertaining to ornithischian dinosaurs. C) Traces likely pertaining to theropod dinosaurs. Dashed lines in B indicate possible track series and left/right pairs. Grid scale = 1 m.

digit lengths, widths of digit bases, heel to digit tip lengths, heel to hypex lengths, divarication angles, length of digit III extension beyond the tips of II and IV (Fig 3).

For trackway sequences—defined as two or more sequential tracks produced by the same individual—additional measurements included (where possible) pace and stride lengths, pace angulation, relative foot rotation, and compass bearing of the trackway midline. Most linear measurements (e.g., track length, track width, stride length) were taken on site using a flexible tape measure to the nearest mm, while orientations were taken using a Brunton compass to the nearest degree. Angular (i.e., digit divarication angles) and additional linear measurements were made using scaled photographs in ImageJ v.1.53.

## Geological setting and site description

The Belly River Group comprises, from oldest to youngest, the Foremost, Oldman, and Dinosaur Park formations [14] that form an eastward thinning clastic wedge of sediment shed from the cordillera to the west during an overall transgression of the Western Interior Seaway [15]. At the Park, the Dinosaur Park Formation is approximately 80 m thick, disconformably overlies the Oldman Formation, and is gradationally overlain by marine shales of the Bearpaw Formation (Fig 3A). Recent U-Pb zircon geochronology of bentonites (i.e., weathered volcanic ash) has constrained the age of the Dinosaur Park Formation as 76.47 + 1.40/-0.084 to 74.44 + 0.30/-0.11 Ma [16]. The Plateau Tuff (75.64 +/- 0.025 Ma) is a regionally extensive bentonite in the middle of the DPF that can be used as a stratigraphic marker [16].

The Dinosaur Park Formation at the Park is characterized by fluvial channel-belt, floodplain, and coal deposits that record sedimentation on an alluvial plain westward of the contemporaneous Western Interior Seaway [17]. Paleochannel

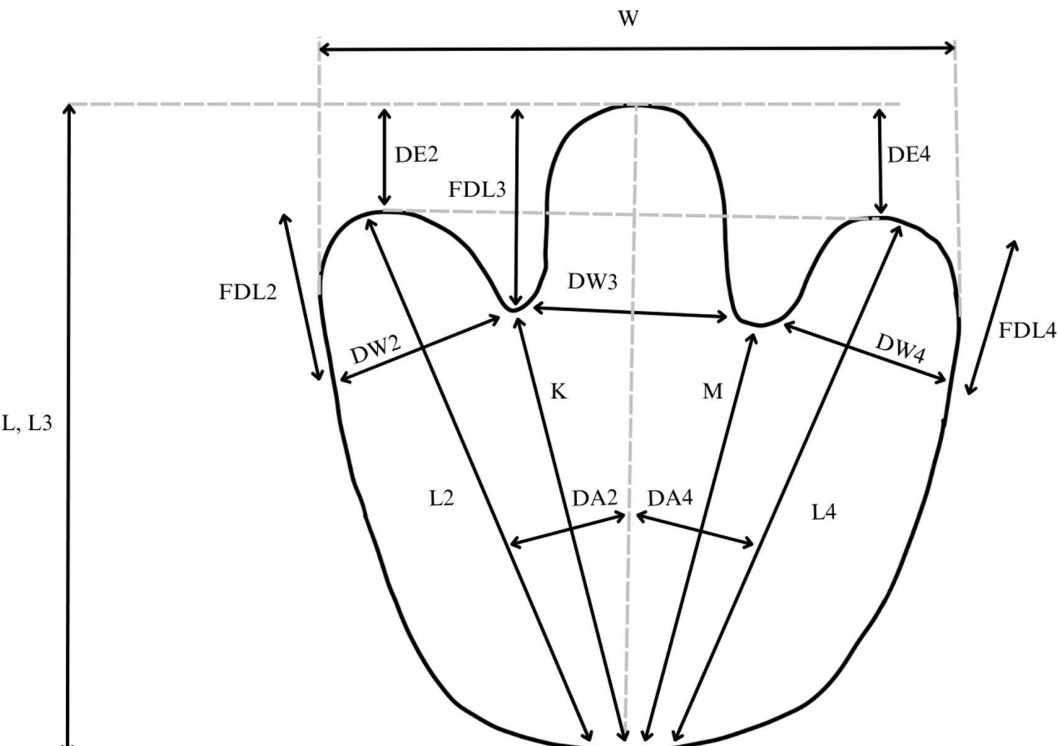

**Fig 3. Diagram showing morphometric measurements taken for each vertebrate trace.** Measurements largely follow Therrien and colleagues [12] and Salisbury and colleagues [13]. Image modified from Therrien and colleagues [12].

reconstructions reveal meandering rivers up to 10 m deep and approximately 200 m wide that constructed single-story channel-belt deposits 10 m in thickness and a few kilometers in width [18,19]. Associated floodplain deposits are dominantly mudstones with thin sandstones and coals that record overbank sedimentation and paleosol development [20].

The Skyline Tracksite (TMP L2467) is located in the upper portion of the Dinosaur Park Formation, stratigraphically above the Plateau Tuff (i.e., younger than 75.64 Ma) (Fig 4A). The elevation of the site is 704.88 m MSL (684.22 m HAE) and ~ 36 m above the contact with the Oldman Formation, which put it within the megaherbivore assemblage zone (MAZ-2a) of Mallon and colleagues [21] (partly equivalent to the *Lambeosaurus-Styracosaurus* dinosaur assemblage zone of Ryan and Evans [22] (Fig 4A). The tracksite is located within a 12-m-thick succession of channel-belt deposits that overlie older floodplain deposits (Fig 4B).

A 10-cm-thick mudstone hosts the trackway, which overlies a 2-m-thick succession of current-rippled sandstone and is overlain by 80-cm of interbedded fine current-rippled sandstone and bioturbated thin mudstone beds. These sediments record moderate-to-low energy deposition in an abandoned channel setting within a fluvial meandering channel-belt environment. The trackway host mudstone bed was likely formed as the result of overbank flooding and suspension settling of

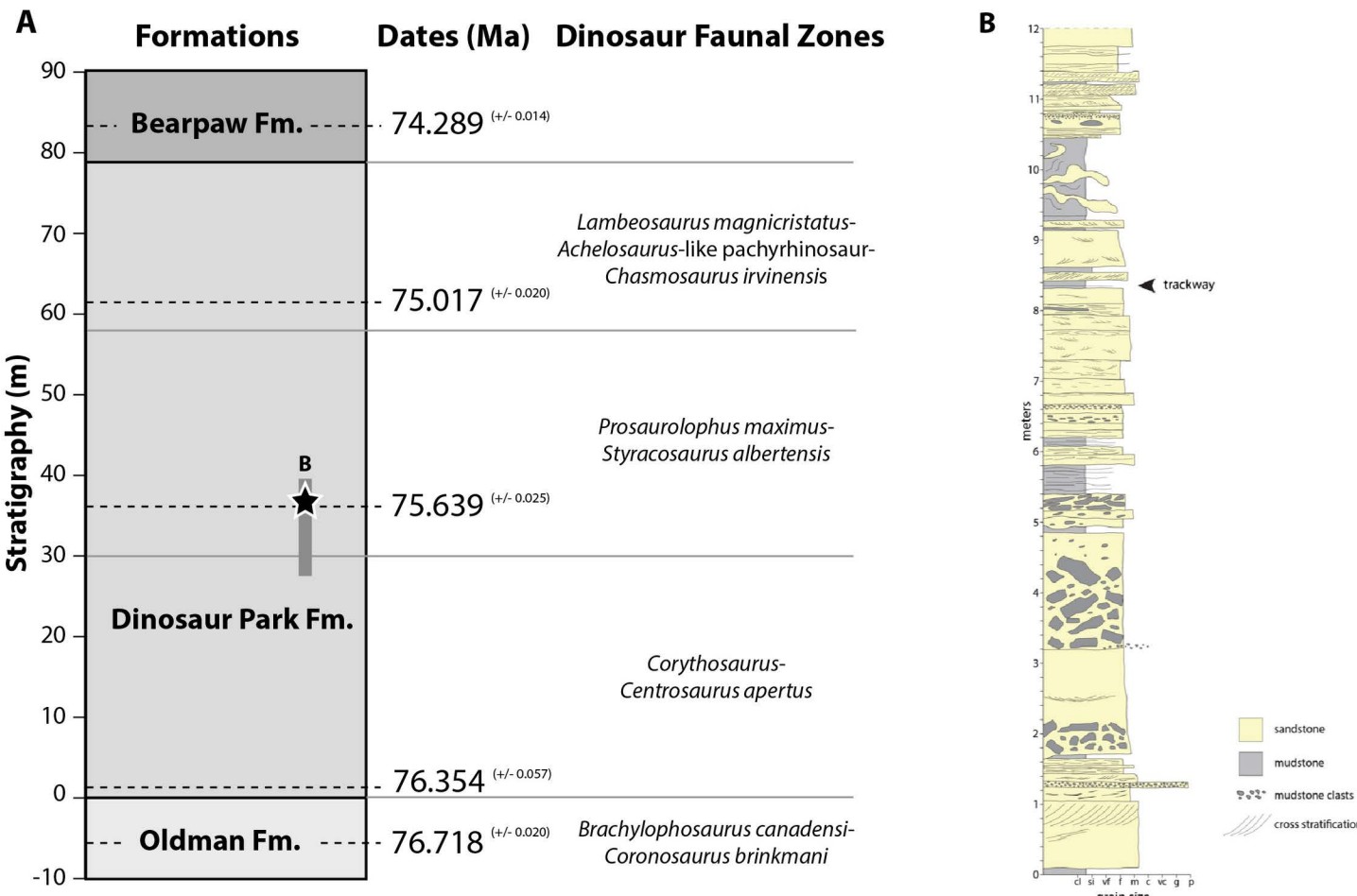

**Fig 4. Geological context for Skyline Tracksite.** A) Regional geological context for Skyline Tracksite (star) within the Dinosaur Park Formation and the *Prosaurolophus maximus-Styracosaurus albertensis* dinosaur megaherbivore faunal zone. Grey bar indicates the extent of the stratigraphic column in B. B) Stratigraphic section through Skyline Tracksite in the Dinosaur Park Formation. Trackway located at 8.4 m above base of section.

fine-grained sediment delivered to an abandoned channel environment. As water levels receded, the mud flat was subaerially exposed in the local topographic low of the abandoned channel. Subsequent moderate-to-low energy unidirectional flow through the abandoned channel environment led to burial and preservation of the trackway. Trace fossils in the overlying sediments are indicative of a quiescent, oxygenated environment consistent with a lacustrine setting contained in an abandoned channel, proximal to the active river channel and episodically flooded. The overall succession of channel-belt deposits exposed at the Skyline Tracksite are consistent with recent paleoenvironmental reconstructions from other study areas within the Park (e.g., [18,19] that reveal a low-gradient meandering river and associated floodplain environment.

Following excavation, the track-bearing horizon measures 13 m long and up to 7 m wide, although the layer continues to the northwest and southeast of the present excavation into the adjacent slopes (Figs 1B, 2A, and 5A).

To the southwest (i.e., into the hill), the sedimentology of the track-bearing horizon grades from strongly indurated and iron-rich to more carbon-rich with abundant plant remains. As a result, the separation between the track-bearing surface and the overlying sandstone is less well developed making it difficult to follow (and excavate) in a southwesterly direction. Unlike the ironstone-rich (northeast) portion of the track-bearing horizon, which, except for the various ichnites, is smooth and featureless on its surface, the organic-rich southwesterly portion preserves common linear ripples (orientation = 102–282°). Unfortunately, current direction could not be measured from these ripples as they were consistently damaged during removal of the overlying beds due to the poor separation between layers. In addition, ichnites were less well defined in the organic-rich zone and had significantly less prominent sediment displacement rims. In contrast, tracks to the northeast show evidence of suction, including relatively deep tracks with steep-sided walls and prominent displacement rims. We interpret these changes in lithology and sedimentary features to reflect a gradient in depositional environments from exposed, subaerial and muddy conditions (in the northeast) to shallow subaquatic but firmer conditions (in the southwest).

Although the purpose of this paper is to document the vertebrate ichnites found on the primary track-bearing horizon, common invertebrate traces (burrows) were found in the overlying mudstone within 30 cm of the track-bearing horizon. Three distinct types of invertebrate traces were present, all preserved as medium-grained sandstone infill within a host matrix. The most common traces are linear, vertical burrows (2–5 mm cross-sectional diameter) (Fig 6C), identified here as *Skolithos* [23]. Less common are larger unbranching, sub-horizontal, U-shaped burrows (5–9 mm diameter; 40–60 mm long) (Fig 6A and B – TMP 2024.012.0321), identified here as cf. *Rhizocorallium* [24].

The majority of the U-shaped-burrows preserve only the infill; however, some preserve *spreiten*-like features between the marginal tubes. While the majority of *Rhizocorallium* occurrences are in marine settings, freshwater occurrences are known [25] and the u-shaped traces also bear broad similarity to sub-recent continental mayfly burrows [26]. Finally, the slightly higher portion of the mudstone with interfingering sandstone horizons, bears smaller, irregular, horizontal burrows (TMP 2024.012.0320), identified here as *Planolites*. Invertebrate traces were, however, not observed in the main track-bearing layer.

## Results – Vertebrate ichnology

### *Ceratopsipes* isp – C1.1, C1.2, C2.1, C2.2, C3.1, C3.2, C4.1, C5.1, C5.2 (Figs. 2B, 5B-C, and 7)

**Description.** A total of nine tracks are assigned to this morphotype and at least eight represent sequential tracks made by four individuals (Fig 2B). Therefore, the tracks represent at least five individuals. All tracks pertain to the pedes; no definitive manus tracks were identified anywhere on the bedding surface. The best-preserved track (C1.1, a right pes) (Figs 5B, 7A and B) is roughly circular in overall dimensions, slightly longer than wide with four broad, blunt digits and a prominent sediment displacement rim surrounding a broadly rounded but asymmetric (laterally skewed) 'heel' (= metatarsophalangeal pad) (Table 1).

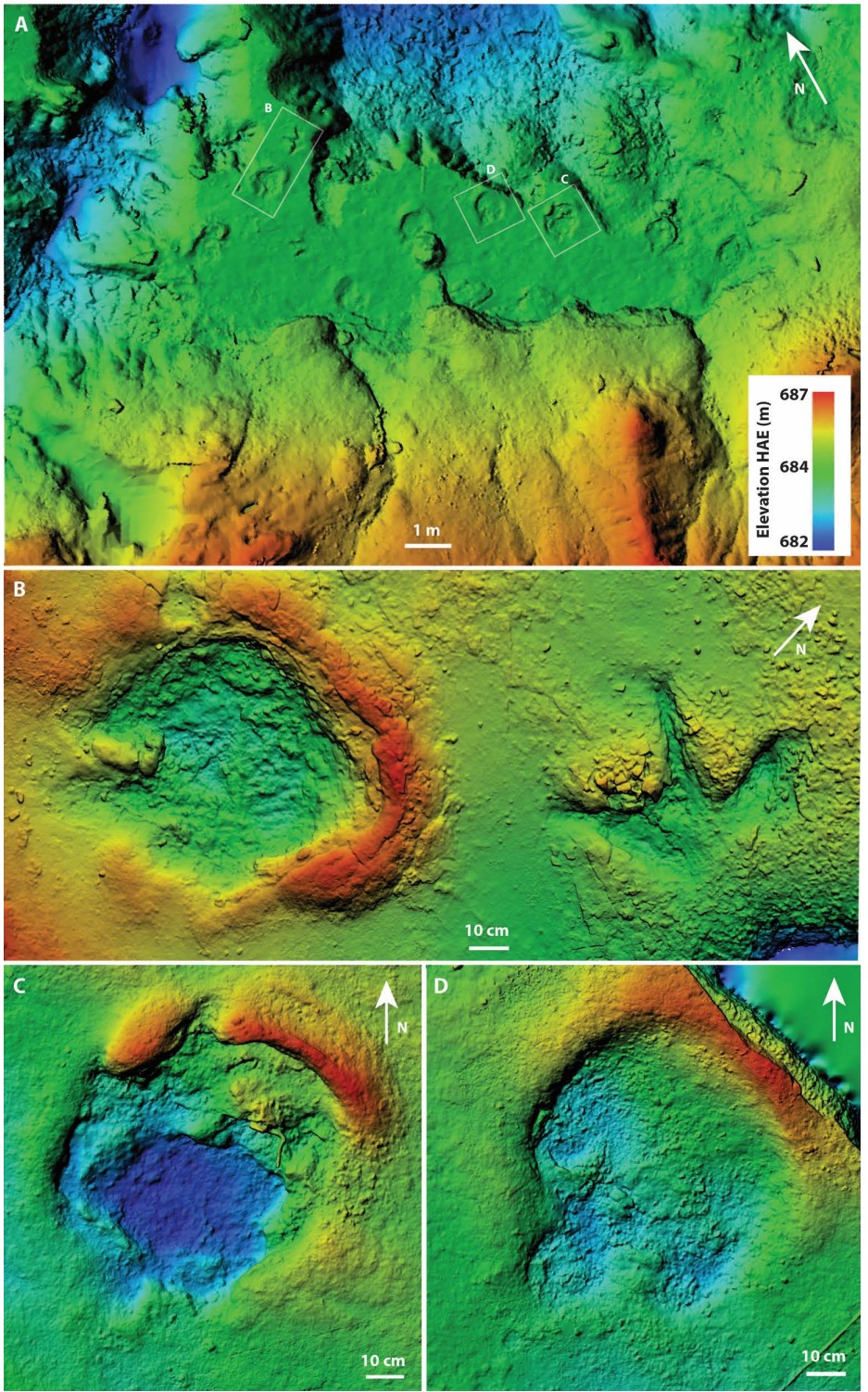

**Fig 5. Photogrammetrically generated digital elevation model with heat map showing elevation of the Skyline Tracksite (A) and select individual prints (B–D).** A) Model of excavated portion (as of 2024) of main trackway, with heatmap showing absolute elevation. Insets show location of B–D. B) Model of two adjacent traces, C1.1 (left) and T1.2 (right), with relative elevation heatmap. C) Model of C3.1, with relative elevation heatmap. D) Model of A1.1, with relative elevation heatmap. All models in plan view. Elevation legend in A, only pertains to A, as B–D are relative elevation only.

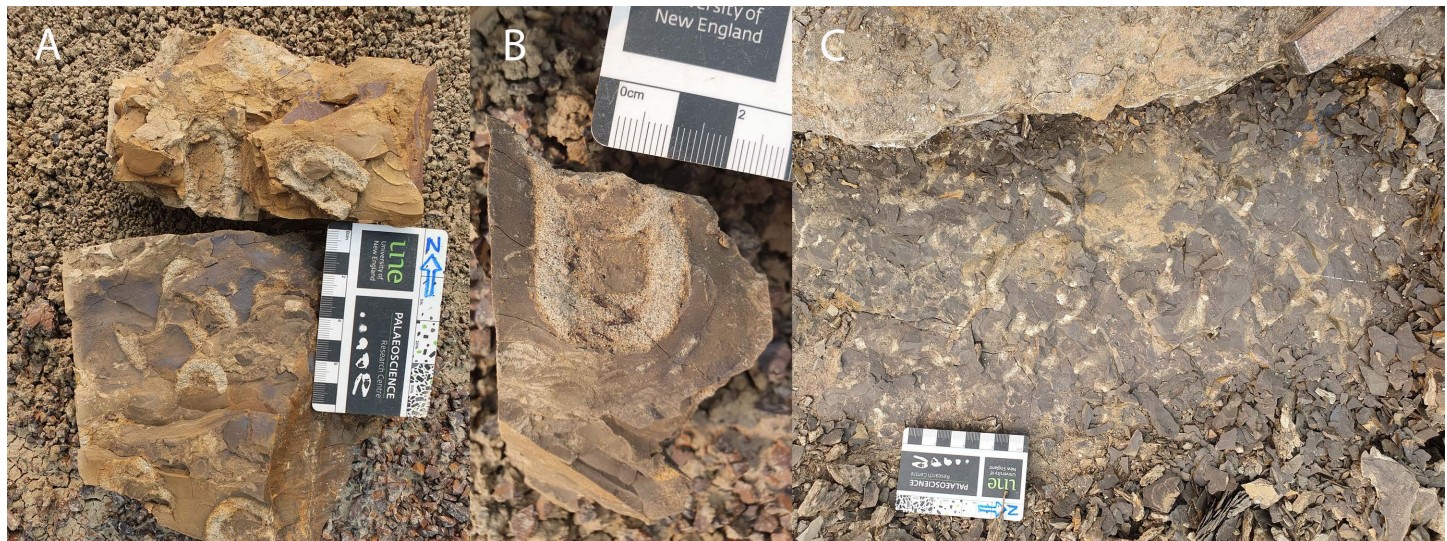

**Fig 6. Examples of the abundant invertebrate trace fossils found in the lower 30 cm of the mudstone overlying the vertebrate trace bearing horizon of the Skyline Tracksite.** A and B) U-shaped marginal tubes of c.f. *Rhizocorallium*, both without (A) and with (B) *spreiten*. C) vertical burrows of *Skolithos.* Scale = 8 cm.

Other tracks of this morphotype typically display a rounded 'heel' and have an overall circular shape but with various ill-defined digit impressions, often with shallow, trough-like toe drag marks (e.g., C1.1, C1.2, C2.1) making precise track length measurements difficult (Fig 7C). For example, C1.2, representing the left pes (and possibly the overprinted manus), is considerably longer than wide with a prominent posterior sediment displacement rim and poorly-defined digits I and II. The lateral digits (digits III and IV) are not discernible; instead, the area where they would be expected consists of a broad drag mark that extends from the anterolateral margin of the track.

**Remarks.** Broadly rounded pedal tracks with four blunt digits are characteristic of both ceratopsid and ankylosaurian trackmakers. Although manual proportions, especially in relation to the pes, are considered an important distinction between these two trackmakers [27], no conclusive manus impressions have yet been found at the Skyline Tracksite, TMP L2467. Therefore, the following discussion concerns only the well-preserved pedal tracks. At the time of writing, only a single ichnotaxon had been erected for both ceratopsid and ankylosaurian tracks: *Ceratopsipes goldenensis* and *Tetrapodosaurus borealis*, respectively [27,28]. Based on the type material of both ichnotaxa, McCrea and colleagues ([27] considered *Ceratopsipes* tracks to be symmetrical with relatively short and proportionally broad digits. In contrast, *Tetrapodosaurus* has a less symmetrical pes, in part due to a short digit I compared to the other digits, although digits are in general proportionally longer, and more slender than in *Ceratopsipes*. Although the holotype pes of *Ceratopsipes* has a more attenuated 'heel' than the present examples from the Park, it is clear that natural variation—ranging from rounded to more V-shaped—exists in successive tracks from the same individual (including the trackway from which the holotype derives) and between other tracks assigned to *Ceratopsipes* [28–30]. The tracks from the Skyline Tracksite TMP L2467 share with *Ceratopsipes* a generally symmetrical pes with four relatively short, broad digits, therefore we assign the present tracks to *Ceratopsipes*, which is widely accepted as a ceratopsid track maker [28]. Absence of manual tracks precludes further specific diagnosis.

Ceratopsids are diverse and common components of the Dinosaur Park Formation, but with discrete biostratigraphic ranges within that formation [21]. TMP L2467 occurs within MAZ-2a, approx. 6 m above and 180 m in a straight line from Bonebed 42, a mixed assemblage dominated by the centrosaurine *Styracosaurus albertensis* [11,31,32]: although *Chasmosaurus belli* [21], as well as potentially *Chasmosaurus russelli* and *Chasmosaurus irvinensis* [33,34], also occur

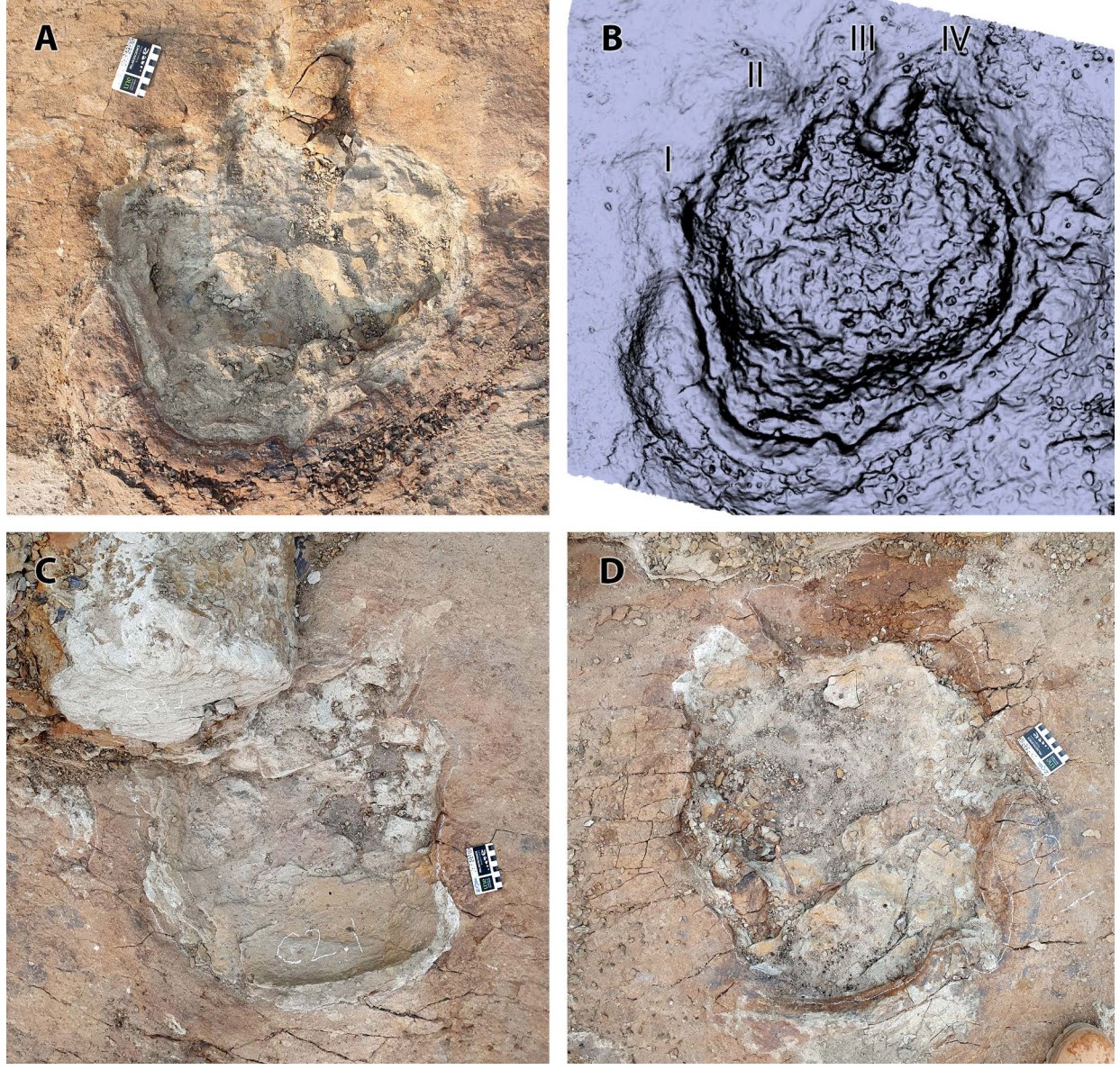

**Fig 7. Individual tracks of *Ceratopsipes* isp.** A and B) trace C1.1 in photograph (A) and digital model (B). C) Photograph of trace C2.1. D) Photograph of trace C4.1. Scale = 8 cm. Roman numerals indicate interpreted digits.

throughout MAZ-2a. Thus, there are at least two potential candidates for the *Ceratopsipes* trackmakers at TMP L2467, the centrosaurine *Styracosaurus albertensis* and a species of the chasmosaurine *Chasmosaurus*.

### Ankylosauria – A1.1, A1.2 (Figs 2B, 5D, and 8)

**Description.** A1.1 is a large tridactyl track tentatively identified as a right based on the reduced (digit IV) and its position relative to a second, partially-excavated track (A1.2) that is interpreted to be part of the same trackway (Figs 2B, 5D, and 8).

**Table 1. Skyline Tracksite footprint data.**

| Taxon | Code | Side | Coordinates | Orientation (°) | Max length (cm) | Max width (cm) | Free digit length (cm) I | II | III | IV | Max digit width (cm) I | II | III | IV | Heel to digit tip (cm) I | II | III | IV | Heel to hypex (cm) I-II | II-III | III-IV | Divarication (°) I-II | II-III | III-IV | Digit extension beyond II and IV (cm) II | Ichnofossil ID | Notes |
|---|---|---|---|---|---|---|---|---|---|---|---|---|---|---|---|---|---|---|---|---|---|---|---|---|---|---|---|
| Ceratopsid | C1.1 | R pes | D.8, 8.1 | 221 | 71 | 60 | 4† | 8† | 12† | 11† | 8† | 9† | 10† | 10† | 45† | 55† | 71 | 62† | 44† | 45† | 47† | 15† | 17† | 18† | 11† | Ceratopsides | |
| Ceratopsid | C1.2 | ? | F.3, 9.2 | ? | 99 | 57 | * | * | * | * | * | * | * | * | * | * | * | * | * | * | * | * | * | * | * | | Morphology unclear; trackmaker may have slipped. |
| Ceratopsid | C2.1 | L pes | E.9, 5.0 | 220 | 75 | 64 | 5.5† | 10† | 13†^ | * | 9† | 11† | 12†^ | * | 48† | 66† | 75 | ? | 48† | 55† | ? | 10† | 19† | ? | ? | Ceratopsides | |
| Ceratopsid | C2.2 | R pes | G.1, 6.5 | ? | 65 | 69 | * | * | * | * | * | * | * | * | * | * | * | * | * | * | * | * | * | * | * | | |
| Ankylosaur | A1.1 | R pes | E.2, 3.7 | 205 | 67 | 61 | na | 13.5 | 19 | 6† | na | 12 | 19 | 11† | na | 55 | 67 | 55 | na | 45 | 52 | na | 24† | 24† | 18 | Ankylosaurid | |
| Ankylosaur | A1.2 | L pes | F.8, 3.8 | ? | ? | ? | * | * | * | * | * | * | * | * | * | * | * | * | * | * | * | * | * | * | * | | Measurements not taken, still partially covered by overburden |
| Ceratopsid | C3.1 | R pes | E.4, 2.4 | 203 | 73 | 74 | 4† | 9† | 14† | 11†^ | 7† | 9† | 11† | 9†^ | 52† | 65† | 73 | 62† | ? | ? | ? | 11† | 17† | 18† | 14† | Ceratopsides | Possible manus print that has been over-printed. |
| Ceratopsid | C3.2 | L pes | F.9, 2.7 | ? | ? | ? | * | * | * | * | * | * | * | * | * | * | * | * | * | * | * | * | * | * | * | | Measurements not taken, still partially covered by overburden |
| Ceratopsid | C4.1 | L pes | E.5, 0.4 | 206 | 64 | 57* | 5.5† | 13† | 14†^ | * | 10 | 11 | 10†^ | * | 54† | 62† | 64 | * | 45† | 47.5† | ? | 15.5† | 17† | ? | ? | Ceratopsides | |
| Ceratopsid | C5.1 | L pes? | F.0, -1.2 | ? | 74 | 67 | * | * | * | * | * | * | * | * | * | * | * | * | * | * | * | * | * | * | * | Ceratopsides | Measurements not taken, requires sediment removal |

*(Continued)*

**Table 1.** (Continued)

| Taxon | Code | Side | Coordinates | Orientation (°) | Max length (cm) | Max width (cm) | Free digit length (cm) | | | | Max digit width (cm) | | | | Heel to digit tip (cm) | | | | Heel to hypex (cm) | | | Divarication (°) | | | Digit extension (cm) | Ichnofossil ID | Notes |
|---|---|---|---|---|---|---|---|---|---|---|---|---|---|---|---|---|---|---|---|---|---|---|---|---|---|II beyond II and IV|---|---|
| | | | | | | | I | II | III | IV | I | II | III | IV | I | II | III | IV | I-II | II-III | III-IV | I-II | II-III | III-IV | II beyond II and IV | | |
| Ceratopsid | C5.2 | R pes? | F.9, −1.5 | ? | ? | ? | * | * | * | * | * | * | * | * | * | * | * | * | * | * | * | * | * | * | * | | Measurements not taken, still partially covered by overburden |
| Tyrannosaurid | T1.1 | L pes | D.8, 4.4 | 300 | 45^ | 51^ | na | ? | 32 | 18 | na | ? | 13 | ? | na | 37.5^† | 45^ | 36† | na | 18^† | 16^† | na | ? | 46 | 21 | Tyrannosaurid | |
| Tyrannosaurid | T1.2 | R pes | D.2, 7.6 | 298 | 45 | 51 | na | 23 | 31 | 20 | na | 12 | 8 | 8 | na | 34† | 45 | 33.5† | na | 19.5† | 13.5† | na | 74 | 33 | 20 | Tyrannosaurid | |
| Tyrannosaurid | T2.1 | L pes | F.2, 7.4 | 290 | ? | ? | na | ? | 32 | 25 | na | ? | 15 | 13 | na | ? | ? | ? | na | ? | ? | na | ? | 31 | ? | Tyrannosaurid | |
| Small theropod-like | ST1 | R pes? | E.8, 1.8 | 233 | 13 | 10 | na | 5 | 7.5 | 5 | na | 1 | 2 | 1.5 | na | 9† | 13 | 8† | na | 5† | 5† | na | 65 | 78 | 8.5† | Small theropod-like | Siding of DII vs DIV uncertain. Tentatively regarded as a right. |

^Approximate due to imprecise preservation of one or more boundaries.

*Incomplete or not completely excavated.

†Estimated using imageJ.

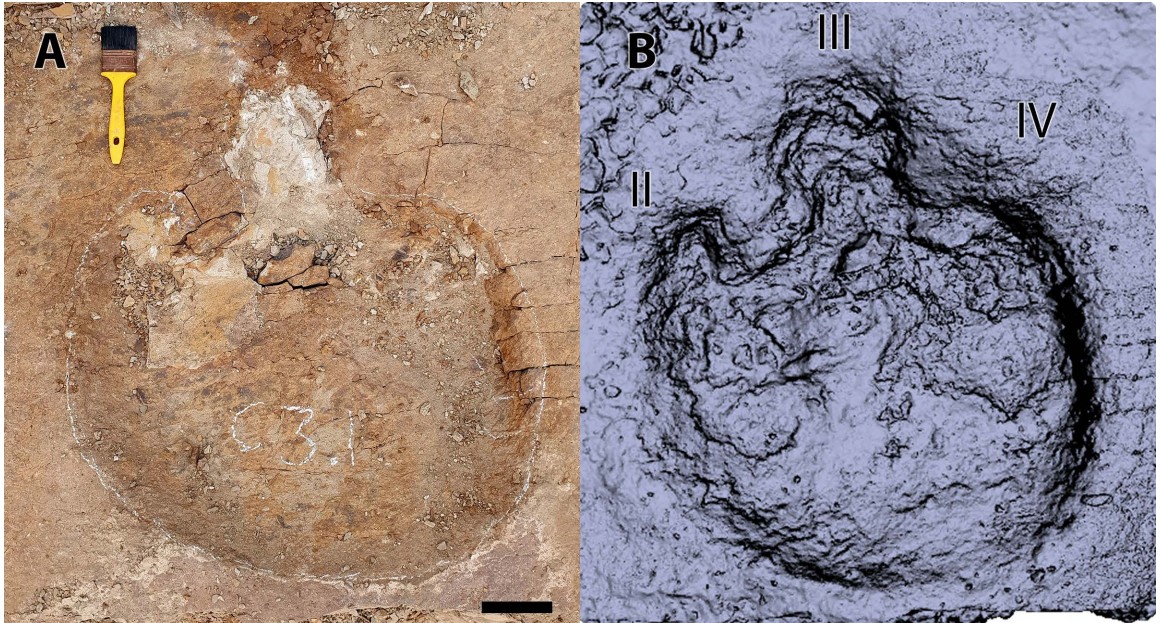

**Fig 8. Single track of possible ankylosaurid (A1.1).** (A) photograph. (B) digital model. Scale = 10 cm. Roman numerals indicate interpreted digits.

A1.1 is well preserved, only slightly longer than wide (67 x 61 cm) and dominated by a large, round, symmetrical metatarsophalangeal pad measuring 45 cm in diameter and constituting ~72% of the total length of the track (Table 1 and Figs 5D and 8). Digit III, the largest of the digits, is blunt and rounded with a prominent proximal constriction. The remaining digits are less well demarcated; digit II is smaller than digit III, with a blunt terminus, a convex medial margin and a straight lateral margin. Owing to the constricted base of digit III, the hypex between digits III and II forms a broad, almost circular concavity in plan view. Digit IV is identifiable only as a weak, blunt prominence on the anteromedial margin of the track. There is no indication of the manus.

**Remarks.** The tridactyl form of A1.1 distinguishes it from ceratopsids and all known ankylosaurid tracks, which have a tetradactyl pes [27,28]. Interestingly, the three blunt digits and symmetrical form resembles some stegosaurian tracks (*Deltapodus* isp., *Garbina* isp., *Stegopodus* isp.); however, there is no evidence of North American stegosaurs after the Jurassic [35]. *Deltapodus* also differs in having pedal tracks that are much more elongate than A1.1 [36]. Whereas tracks of *Stegopodus* are asymmetrical (as a result of posterior extension of a proximal pad at the base of digit IV) and have digits that barely project beyond the hypex [37], A.1.1 is notably more symmetrical with a rounded posterior track margin, and has digits that are comparatively elongate (especially digits III and?II). *Garbina*, originally named for Early Cretaceous thyreophoran tracks from Australia [13], shares with A1.1 large (up to 80 cm in *Garbina*) mesaxonic tracks with three blunt digits. However, A1.1 differs in having a digit III with a strong basal constriction, giving it a more rounded (rather than anteriorly tapering) outline. Hadrosaurids are ubiquitous components of the Dinosaur Park Formation fauna and their tracks have been reported previously [3,5]. Although variations naturally exist due to variable substrate properties, kinematics, and preservation, Late Cretaceous hadrosaurid tracks (*Hadrosauropodus* isp.) have a tridactyl pes with blunt digits resembling A1.1; however, they tend to be wider than long with longer free digits that are ovoid, the long axes of which are oriented parallel to the track axis, and have rounded or bilobed heel impressions, with metatarsophalangeal pads that occupy, on average, 52% of the proportion of the total track length [38–40]. In contrast, A1.1 is longer than wide, with relatively small, short, rounded digits and a rounded metatarsophalangeal pad that occupies a greater portion

(72%) of the total length of the track. Although ankylosaurians (including *Tetrapodosaurus* isp.) are normally associated with a tetradactyl pes, a number of Late Cretaceous ankylosaurids have a tridactyl pes, including *Euoplocephalus tutus*, *Dyoplosaurus acutosquameus* and *Pinacosaurus grangeri* [41–43]. Notably, both *Eupolocephalus* and *Dyoplosaurus* come from the Dinosaur Park Formation: *Euoplocephalus* occurs throughout MAZ-2a, whereas *Dyoplosaurus* is known from only two occurrences in MAZ-1 [41]. A third taxon, the nodosaurid *Panoplosaurus mirus*, occurs throughout MAZ-1 [21]. Although the pes is unknown in *Panoplosaurus*, there is some indication that nodosaurids possessed tetradactyl pedes [42], which would potentially distinguish them from the tridactyl track A1.1. All previously-described ankylosaurian tracks have been attributed to the tetradactyl ichnotaxon *Tetrapodosaurus* [27], which is unsurprising given that the majority of these tracks come from 'middle' Cretaceous (Aptian–Turonian) strata [4,27,44]. By comparison, few Late Cretaceous ankylosaur tracks have been reported where tridactyl forms might conceivably exist [45,46]. However, McCrea and colleagues [4] noted ankylosaurid tracks with tridactyl pedes in the Cenomanian Dunvegan Formation of British Columbia. These tracks were not described but were said to bear "some resemblance to Sternberg's [47] *Tetrapodosaurus* ichnosp." [4] and noted as having five manual digit impressions but only three pedal digit impressions [4]. A comparison between the only published image of these tracks [4], Fig 8) and A1.1 shows distinct similarities, especially in the overall rounded form of the pes with three blunt digit impressions and one digit (digit II?) that appears smaller and less developed than the remaining two.

After our article was accepted for publication, the tridactyl ankylosaurian tracks originally reported by McCrea and colleagues [48] were described as a new ichnotaxon, *Ruopodosaurus clava*, by Arbour and colleagues [49]. Those tracks and additional assigned material come from the Cenomanian-aged Dunvegan and Kaskapau formations in British Columbia, with the exception of one specimen, TMP 1994.183.0001, from the Dunvegan Formation in northwestern Alberta. Arbour and colleagues [49] diagnosed *Ruopodosaurus clava*, in part, by having pes tracks that are weakly mesaxonic with rounded or bilobed heel, three robust digits ending in blunt triangular or U-shaped toe tips, and divarication between digits II and IV between 60–80°. This description generally matches that of A1.1 reported here although digit divarication could not be measured in A1.1 due to the short, indistinct impression of digit IV. Other diagnostic features of *Ruopodosaurus clava* related to trackway parameters and manus track morphology are not presently available from the tracks reported here. Nevertheless, based on the distinctive tridactyl configuration of A1.1, we conservatively assign it to cf. *Ruopodosaurus*.

**Tyrannosauridae – T1.1, T1.2, T2.1 (Figs 2C, 5B, and 9)**

**Description.** Three large (~45 cm long) tridactyl theropod tracks were identified, two of which (T1.1 and T1.2) pertain to a partial trackway (Figs 2C, 5B, and 9).

Based on the medial curvature of digit III, and a more posteriorly-situated hypex between digits III and IV compared to that of digits II and III [39,48], T1.1 is identified as a left and T1.2 is a right. However, given that a significant portion of the track-bearing horizon is missing and the great distance (3.37 m) between the two tracks, at least two intervening tracks appear to be missing from the sequence (Fig 2C). Both T1.1 and T1.2 are tridactyl, mesaxonic with long, tapering digits that terminate in sharp ungual impressions (Fig 9). The claw trace of digit III in T1.1 is medially curved, whereas that of T1.2 is more symmetrical. There is no trace of a hallux. Digit II and III form a hypex that is close to 90° in T1.2, whereas digit II of T1.1 is only visible as shallow depression. The walls of the digits, especially digits III and IV, which are most deeply impressed, are steep-sided. The metatarsophalangeal impression is relatively small compared to overall track length (free digit length of digit III:track length = 0.7). The 'heel' of T1.2 tapers to an obtuse V-shaped angle, the apex of which is roughly in line with the long axis of digit III; the posterior border of T1.2 is less distinct and cannot be compared to T1.2. As best seen in T1.2, there is no visible medial notch at the base of digit II (an otherwise common theropod track feature), but this may be due to the poorly-imprinted posterior track margins of most tracks.

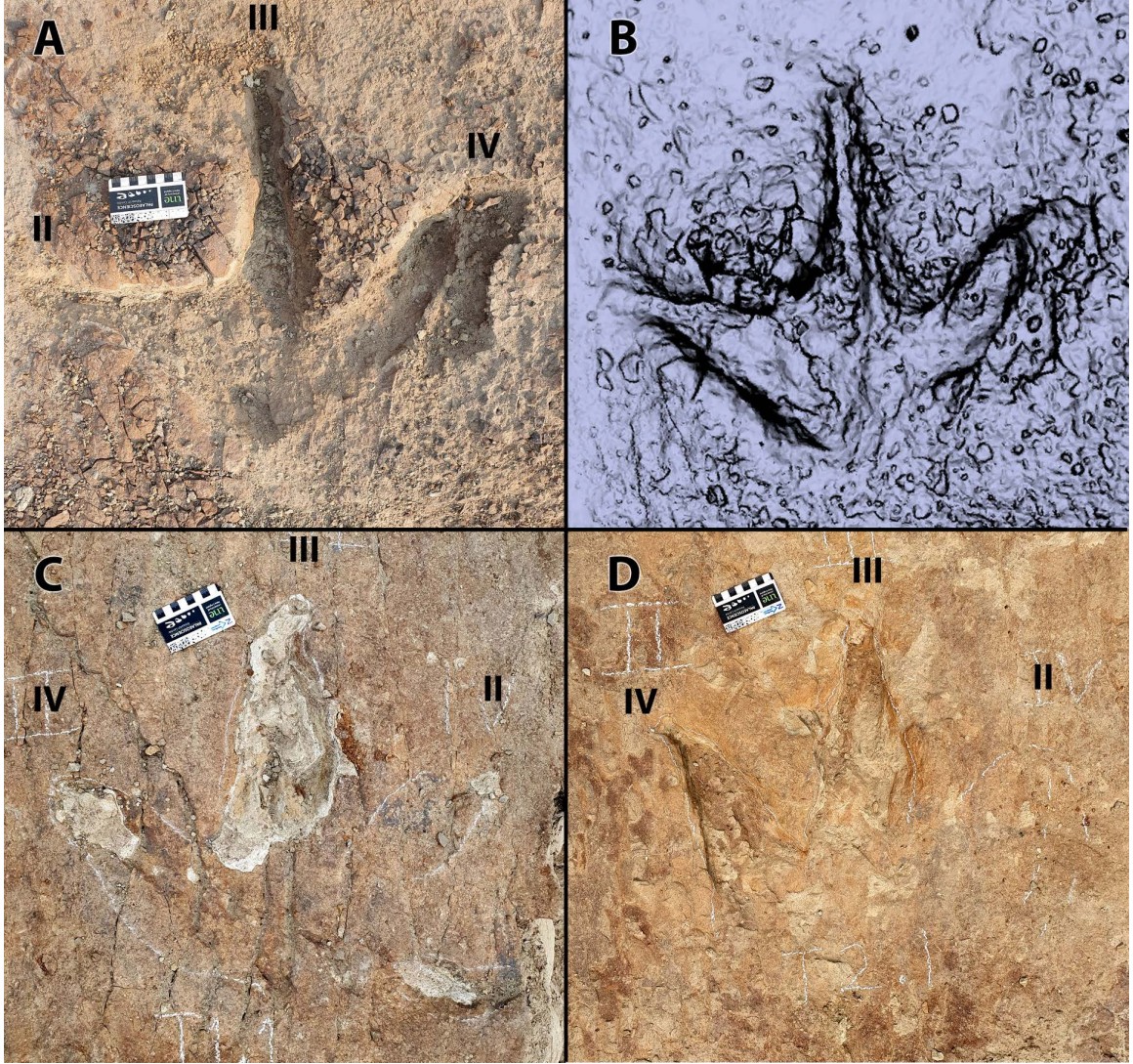

**Fig 9. Individual tracks of tyrannosaurids.** A and B) trace T1.2 in photograph (A) and digital model (B). C) Photograph of trace T1.1. D) Photograph of trace T2.1. Scale = 8 cm. Roman numerals indicate interpreted digits.

The third trace, T2.1, is moderately well preserved and represents the passage of a second individual; track orientations reveal both animals were travelling in a northwesterly direction (Table 1), in approximately parallel paths. Based on comparisons with T1.1–2, T2.1 is identified as a left pes. Digits III and IV are well defined but both the 'heel' and the entire digit II are poorly impressed and barely perceptible. Both digits III and IV are broad at their base and taper distally into sharp ungual impressions. Unlike in T2.1, the ungual of digit III does not curve medially but is more symmetrical.

**Remarks.** Potential trackmakers of the tridactyl theropod tracks include tyrannosaurids and possibly ornithomimids. Common ornithomimids in the Dinosaur Park Formation include *Struthiomimus* and *Ornithomimus*, although an indeterminate large-bodied ornithomimid, similar in proportions to the Mongolian taxon *Gallimimus,* has also been identified [50]. Based on the scaling of complete ornithomimid pedes, Enriquez and colleagues [51] determined the foot length of largest ornithomimids from the post-Cenomanian of Alberta (i.e., the large-bodied taxon described from

the Dinosaur Park Formation by Longrich [50]) to be approximately 40 cm. Those authors therefore considered tridactyl theropod tracks ≥45 cm long to be too large to be produced by any known ornithomimid [51]. At ~45 cm in length, the large theropod tracks from TMP L2467 fall precisely on this cutoff, suggesting they too were made by tyrannosaurids. Tyrannosaurids from the Dinosaur Park Formation include the tyrannosaurine *Daspletosaurus* sp*.* and the more common albertosaurine *Gorgosaurus libratus* [52]. As their taxonomic distinction is based solely on osteological characters [52,53], it is not possible to distinguish between them based only on tracks; we therefore consider the large theropod tracks at TMP L2467 to have been made by indeterminate tyrannosaurids.

There are currently two named ichnotaxa from the latest Cretaceous that unquestionably represent tyrannosaurid track-makers: *Belloratipes fredlundi* and *Tyrannosauripus pillmorei* [48,54]. McCrea and colleagues [48] diagnosed the tracks of *Belloratipes fredlundi* (in part) as mesaxonic, tridactyl, longer than wide, with broad digits that lack digital pads, and a large metatarsophalangeal pad compared to overall track length (free digit length of digit III:track length = 0.51; [48]). While most of these features apply to the theropod tracks from TMP L2467, the digits are proportionally longer and narrower with a small metatarsophalangeal pad (free digit length of digit III:track length = 0.7), which would argue against an assignment to *Belloratipes*. The tracks of *Tyrannosauripus* are similar to, but larger than *Bellotoripes* (40% longer, 20% wider), have faint digital pad impressions and a medially directed (as opposed to anteriorly directed) hallux [48,54]. Thus, the theropod tracks from TMP L2467 cannot be attributed to *Tyrannosauripus.* In a broad sample of tyrannosaurid tracks from the Late Cretaceous of North America, Enriquez and colleagues [51] found that the free digit length of digit III is a function of absolute size: smaller tracks have a proportionately longer digit III than larger tracks. Given that the type series of *Belloratipes* represent larger tracks ($\mu\ TL = 60$ cm) than those described here ($TL = 45$ cm)—*Tyrannosauripus* are larger still—, we would not expect them to share similar proportions. Instead, the TMP L2467 sample more closely resembles similar-sized

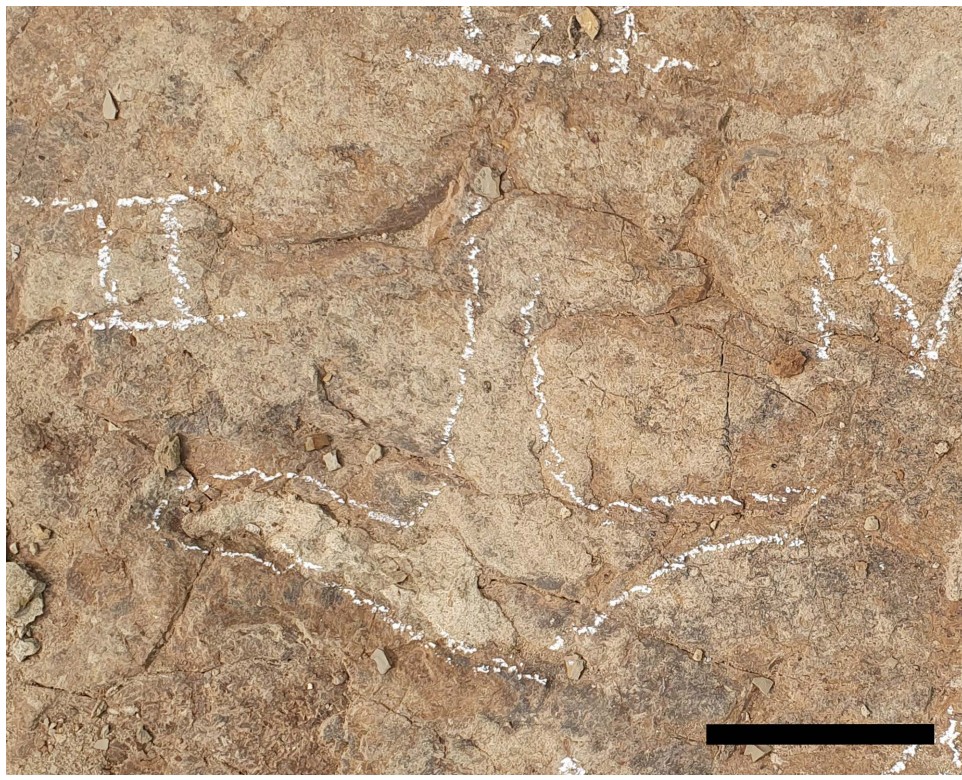

**Fig 10.  Single potential small theropod footprint (ST1) from the Skyline Tracksite.** Scale = 5 cm.

($TL = 40 - 49$ cm) unnamed tyrannosaurid tracks from the partly-coeval Wapiti Formation in central-western Alberta [39,51] especially in having relatively long and narrow digits and a small metatarsophalangeal pad. We therefore refrain from assigning the present tracks to any existing ichnotaxon, preferring to leave them in open nomenclature.

**Small theropod-like taxon ST1 (Fig 10)**

**Description.** ST1 is an isolated and shallowly-impressed tridactyl trace that was excavated and documented in July 2024 (Fig 10) prior to photogrammetry, but because of the friable surface and low track relief had already eroded and was mostly unrecognisable by August of the same year.

As it was isolated, it is difficult to identify as either a right or left, although we treat it tentatively here as a right pes for the sake of the description. It is only slightly longer than wide (TL = 13 cm; TW = 10 cm), strongly mesaxonic, with long and tapering digits (Table 1). Divarication angles are relatively high (II^III = 65; III^IV = 78). Digit III is the longest digit and is straight, whereas digits II and IV are shorter, subequal in length, and curved along their long axis in a posterior direction; the curvature of digit II is stronger than that of digit IV. The metatarsophalangeal pad is moderately large (free digit length of digit III:track length $\mu$ = 0.58), terminating in a broad V-shaped apex that is laterally offset with respect to the long axis of digit III. Digital pads are not discernible and there is no indication of a 'theropod' notch at the base of digit II.

**Remarks.** Relatively small tridactyl tracks with narrow digits are typically regarded as made by either small theropods or birds. A checklist of features for identifying avian traces given by Lockley and colleagues [55] includes: (1) resemblance to modern bird prints; (2) relatively small size (typically < 10 cm); (3) indefinite phalangeal pad impressions; (4) wide total digit divarication angle (II^IV = 110–120° or more); (5) a posteriorly-facing hallux; (6) thin claws; (7) claws on digits II and IV curving away from digit III, and; (8) narrow digits relative to track length [55–57].

In contrast, tracks of non-avian theropods are regarded as having: (1) unequal lengths of digits II and IV, with digit IV being longer than digit II; (2) lower total digit divarication angle (II^IV < 90°); (3) a notch in the medial margin of the metatarsophalangeal pad close to the base of digit II, and: (4) medially curved unguals of digits II and III and a laterally curved ungual of digit IV [58]. Although footprint splay (FL/FW) has sometimes been considered, Xing and colleagues [57] found no evidence that this value differs between large avian and small (non-avian) theropod tracks. Because ST1 is an isolated and imperfectly-preserved trace, many of the above criteria are difficult to observe or to state with certainty (e.g., presence/curvature of unguals, presence of digital pads). Moreover, there is frequently considerable overlap in these features between non-avian theropods and birds either as a result of substrate consistency, biology or other features. For example, although total digit divarication of ST1 (II^IV = 143°) falls within the avian 'range', Xing and colleagues [57] found total divarication values of up to 120° in non-avian theropod tracks and cautioned against using this parameter as the sole distinguishing feature between possible trackmakers. Compounding this problem, Enriquez and colleagues [39] also considered thescelosaurids—which are known from the Dinosaur Park Formation [59] as capable of making small theropod-like tracks, although the tracks they described had lower divarication angles (II^IV = 57–87°) and comparatively broader digits. We also consider small-bodied pachycephalosaurids, which likely had functionally tridactyl pedes [60] and were a diverse and reasonably abundant component of the Dinosaur Park Formation fauna (e.g., [22,61], as potential candidates. We therefore follow the conservative approach of Enriquez and colleagues [39] in referring ST1 to a small theropod-like trackmaker.

## Discussion

Despite a rich history of exploration and unmatched abundance of body fossils, vertebrate ichnology in Dinosaur Provincial Park is still in its infancy [3]. Until relatively recently, only two isolated track casts had been collected (a hadrosaurid [TMP 1981.034.0001] and a tyrannosaurid [TMP 1993.036.0282]) although additional tracks and several badly eroded trackways were reported [2–4]. Following these earlier reports, Therrien and colleagues [5] documented two hadrosaurid trackways (including one first reported by McCrea [3]), highlighting the unusual preservation of concretionary tracks

and forming a valuable new search image for footprints and trackways that has helped to recognise new trackways in other formations with similar geomorphology (e.g., Horseshoe Canyon Formation; [40]). Because of the steep exposures, 'popcorn' regolith, and rapid rates of erosion (2–4 mm/year; [6]) that typify the badlands of Dinosaur Provincial Park [6,7], flat bedding surfaces are rarely exposed and, where present, localised depressions (such as dinosaur tracks) are rapidly filled by sediments. As a result, 'typical' tracks (natural moulds and casts) are unlikely to be recognised. Narrow ironstone benches, which are naturally resistant compared to the surrounding sandstones and mudstones, are common geomorphological features in the Park, yet localised depressions (including mouldic footprints) on these benches are invariably filled with sediments (either by the overlying track infill or from colluvial runoff), obscuring potential tracks. Tracks in the ironstone track-bearing layer at TMP L2467 were initially recognised only because the extrusive sediment rim of a single track was exposed above the overlying sandstone surface and only systematic excavation revealed the rest of the track. During the course of this study, however, we found that displacement rims could be readily identified on nearby ironstone ledges when they protruded above the overlying sandstone and that they could be easily excavated to reveal complete tracks. We therefore predict that 'typical' tracks, like those at the Skyline Tracksite, are much more common within the

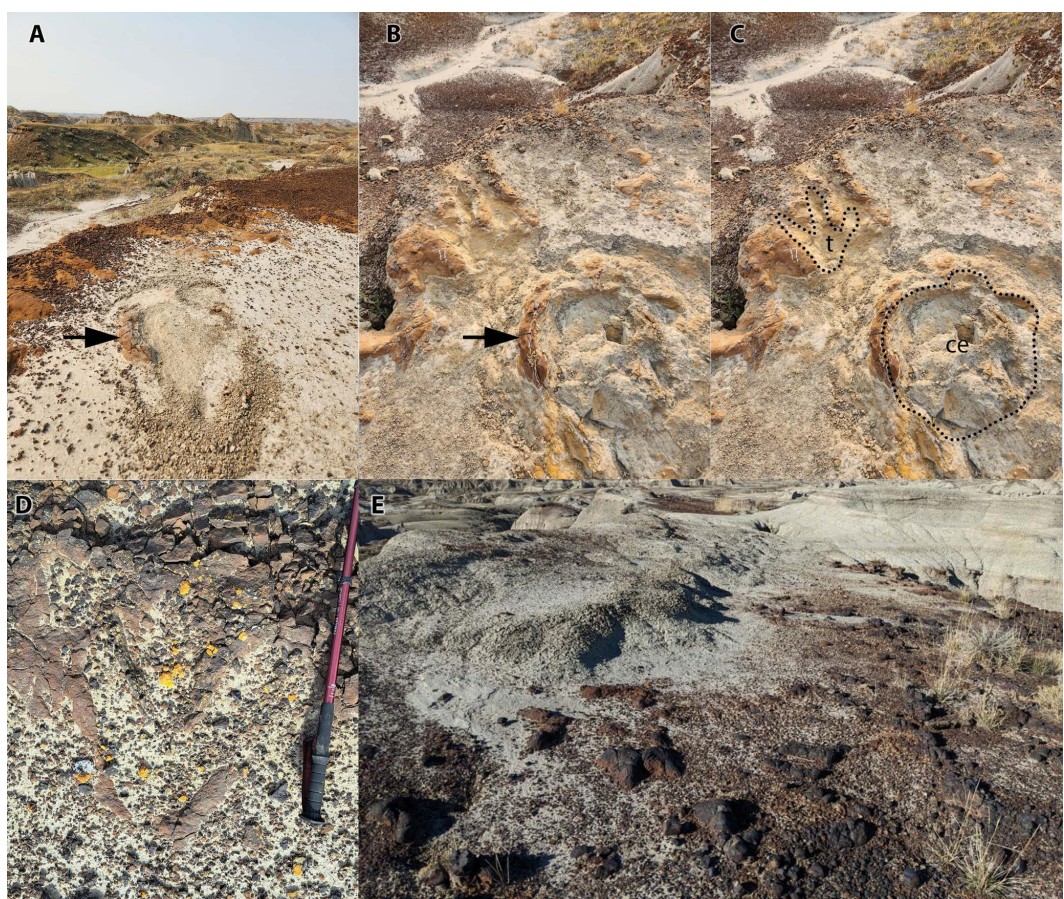

**Fig 11. Additional trackways found in the Dinosaur Park Formation subsequent to Skyline Tracksite discovery.** A–C) 'Cathedral Tracksite' after initial discovery with minimal excavation (A) Sept 9, 2024, and following moderate excavation, Sept 12, 2024 (B), with track interpretation (C) showing a probable ceratopsian track (ce) and a large theropod track (t). Black arrows in A, B indicate the same sediment displacement rim for reference. D) Unexcavated isolated tridactyl print ('Mueller Tracksite' Aug 11, 2024, photo credit Andre Mueller). E) Unexcavated potential trackway ('Lambert Tracksite', Oct 25, 2024, photo credit Dawson Lambert).

area of DPP than previously assumed. Indeed, a brief exploration of the badlands near the Skyline Tracksite in August 2024, as well as ongoing fieldwork in the Park, have led to the discovery of several new tracksites by one of us (BJP) using this new search image (Fig 11). Notably, new tracksites have now been found in both lower and upper parts of the Dinosaur Park Formation, and in varying local depositional environments, suggesting the occurrence of the Skyline Tracksite is not unique.

Once exposed, however, the thin ironstone layers preserving such tracks are prone to rapid erosion (similar to that observed on the concretionary ironstone tracks; [5]), likely influenced by the expansion and contraction of the bentonite-rich sediment layers during temperature and moisture extremes. Hence, once exposed at the surface, trackways such as the ones described here are unlikely to remain intact unless discovered and protected. Nevertheless, this new search image has the potential to dramatically alter the field of vertebrate palaeoichnology in the Park and for similar formations where badlands occur (e.g., Horseshoe Canyon and Milk River formations).

The discovery of the first concave epirelief tracks and trackways within Dinosaur Provincial Park reveals a snapshot of a Late Cretaceous multitaxic assemblage dominated by dinosaurs. We interpret the site as most likely representing an abandoned meandering channel, characterized by a lacustrine setting with episodic flooding. The environment would have provided the necessary conditions for footprint preservation and explains the presence of worm burrows typically associated with marine environments in the sedimentary layers above the tracksite. Although concretionary tracks attributed to hadrosaurs and a single tyrannosaurid track have previously been found, the tracksite reported here represents a more unusual assemblage including multiple individual ceratopsians, a possible ankylosaurian, at least two tyrannosaurids, and a small theropod-like taxon. Given the abundance of ceratopsids in uppermost Cretaceous strata of North America, it is surprising that relatively few of their tracks have been reported (e.g., [4,28,30,45,62]). Although it is impossible to tell the relative timing between the formation of the various tracks, several lines of evidence imply some level of synchronicity and, therefore, gregariousness or herding behaviour among *Ceratopsipes* trackmakers. In particular, the similar orientations, uniform spacing between individuals, and similar track characteristics (e.g., extrusive rims, slide marks) suggest the *Ceratopsipes* tracks were made at a time when sediment consistency was the same across much of the site, probably by a herd of animals traveling in the same direction. The *Ceratopsipes* tracks are furthermore oriented roughly perpendicular to and heading towards the ripple-laminated surface along the southwestern edge of the quarry, which indicates that these animals were moving towards the waterline, perhaps to drink. Taphonomic analyses of multiple extensive monodominant ceratopsid bonebeds that occur within the Park (including the nearby *Styracosaurus* bonebed [Bonebed 42]) provide strong evidence for gregariousness in this clade [63–66], specifically Centrosaurinae, a conclusion that is mirrored for ceratopsid bonebeds elsewhere across North America (e.g.,[66–69]). However, it is acknowledged that such herding behaviour may only have occurred temporarily or at certain times of the year [66]. Nevertheless, it is not unexpected that the ceratopsids at the new tracksite may have been engaging in similar herding behaviour. If this truly does represent a ceratopsian herd, then *Styracosaurus albertensis* may be the more likely trace maker as there is abundant evidence for gregariousness in this taxon in the Dinosaur Park Formation, but not for *Chasmosaurus*.

Similarly, the tyrannosaurid tracks indicate two individuals walking adjacent to one another but parallel to the waterline (rather than towards it, as is the case for the ceratopsid tracks). Uniform orientations and similar track characteristics (e.g., extrusive sediment margins, especially at the hypices; poorly-impressed digit IV) again suggest the tracks were made relatively synchronously (but likely at a different time than the *Ceratopsipes* tracks). Possible gregariousness in North American tyrannosaurids has been interpreted previously from bonebed [70–72] and trackway evidence [48,73], although, as with ceratopsids, the longevity of such gregarious associations may have varied [71]. While such an interpretation of the tyrannosaurid tracks at the Skyline Tracksite could be made, it is far from conclusive.

The absence of manus impressions from either the ceratopsids or the presumed ankylosaurid is perplexing given that both are obligate quadrupeds. Trackways that favour the preservation of either the manus or pes (collectively referred to as either manus-dominant [MDTs] or pes-dominant trackways [PDTs], respectively; [74]) are a well-known phenomenon among sauropods. Whereas MDTs are always the result of the pes failing to leave an impression, PDTs are more complex and may form either from the failure of the manus to leave an impression or as a result of overstepping by the hindfoot, whereby the manus print becomes partially or totally obscured [74]. Although, to our knowledge, neither MDTs nor PDTs have been reported for ankylosaurids or ceratopsids (see [37] and [13] for instances of PDTs among presumed stegosaurian trackmakers), similar reasons as those posed for sauropod PDTs should also apply to these taxa. In their analyses of sauropod tracks, Falkingham and colleagues [74] proposed that animals with a posterior centre of mass (CM) would be more likely to produce PDTs. Although both ankylosaurians and ceratopsids had a CM that was placed well anterior to the acetabulum—as would be expected of a quadruped—it was closer to the hindlimb than to the forelimb, suggesting a greater distribution of weight on the pedes than on the mani [75,76].

Finally, if our interpretation of the ankylosaurian track is correct, its similar alignment, spacing, orientation and preservation relative to the *Ceratopsipes* tracks could suggest the ceratopsians and ankylosaur were moving together in a single herd. In modern day ecosystems, mixed-species groups and aggregations are common and occur regularly in assemblages of fish, birds, large mammals, amphibians, and invertebrates [77]. Individual fitness can be enhanced through dilution benefits (reduced probability of being attacked by a predator; e.g., [78]) or detection benefits (increased probability of identifying a predator; e.g., [79]). Evidence is accumulating that access to social information [80], in this case from heterospecifics [81], appears to be a major driver of this phenomenon [82].

The most useful modern-day comparator for ornithischian macro-herbivores is probably large grazing mammals, and there are many studies on mixed-species assemblages on the African plains. A major explanation for the preference of some species for mixed-species groups appears to be as a defence strategy against shared large predators [83,84], with individuals benefitting from the varied modes of sensory perception of heterospecifics, either by listening to their alarm calls [85] or by monitoring cues such as body posture [86]. This allows individuals to spend less time performing vigilance behaviours [87]. Interestingly, in seasonally wet and dry ecosystems, multi-species herding is reduced during the dry season and increases in the wet season, suggesting that this behaviour is influenced by resource (water) availability [88]. The tyrannosaur tracks within the Skyline Tracksite indicate that large predators shared by ankylosaurs and ceratopsians are present, which corresponds to a key driver of multi-taxic herding behaviour in modern day large mammal herbivores (increased apex predator density; [84,85,87]).

## Conclusions

The Dinosaur Park Formation in southern Alberta preserves one of the richest Late Cretaceous terrestrial communities in terms of the abundance and diversity of its vertebrate fossils. Yet despite more than a century of intensive collecting in the area of Dinosaur Provincial Park and the recovery of substantial numbers of vertebrate body fossils, fossil footprints and trackways have been elusive. While the steep, rapidly-eroding badlands geomorphology makes the Dinosaur Park Formation and related formations in the Western Interior of North America ideal for revealing body fossils, the absence of large planar surfaces (where tracks and trackways would be expected) has led to the widespread assumption that such ichnofossils are generally not preserved. Developing appropriate search images to find new fossil localities is an essential task for field-based palaeontology, and the recognition of peculiar concretionary tracks was an important development for prospecting in the badlands of southern Alberta. The new search image (protruding, often iron-rich sediment displacement rims) developed during the course of this study has already resulted in the discovery of several additional tracksites that are awaiting study. These preliminary findings indicate that vertebrate footprints are far more abundant in badland settings than has been previously accepted. The Skyline Tracksite described here underscores the importance of trackway data that can be gleaned even from well explored formations, such as the Dinosaur Park Formation.

## Supporting information

**S1 Models.  3D Model Gallery.** https://mq.pedestal3d.com/r/bePfWqNGay/.
(DOCX)

## Acknowledgments

We thank Harris Ahmed, LJ Moss, Johana Simonova, and Jamie Stones (University of Reading) and Brady Holbach, Dawson Lambert, Urgon Snider, and Christopher West (RTMP) for assistance with site preparation. Dawson Lambert (RTMP) and Andre Mueller (Alberta Parks) found new tracksites following the discovery of TMP L2467. Jenifer Blacklaws and other Alberta Parks staff provided logistical support for fieldwork. We thank Nathan Enriquez (UNE) for insightful discussions. Veronica Sew (Cansel) provided technical assistance with differential GPS correction. This work took place in the province that is known as Alberta. We gratefully recognize and honour Indigenous peoples' long-standing connections to these lands. This is the second paper arising from the International Palaeoecology Research Field Course led by BJP, CMB and PRB.

## Author contributions

**Conceptualization:** Phil R. Bell, Brian J. Pickles, Caleb M. Brown.

**Data curation:** Phil R. Bell, Brian J. Pickles, Sarah C. Ashby, Issy E. Walker, Sally Hurst, Michael Rampe, Paul Durkin, Caleb M. Brown.

**Formal analysis:** Phil R. Bell, Brian J. Pickles, Paul Durkin, Caleb M. Brown.

**Funding acquisition:** Phil R. Bell, Brian J. Pickles.

**Investigation:** Phil R. Bell, Brian J. Pickles, Sarah C. Ashby, Issy E. Walker, Sally Hurst, Paul Durkin, Caleb M. Brown.

**Methodology:** Phil R. Bell, Brian J. Pickles, Michael Rampe, Paul Durkin, Caleb M. Brown.

**Project administration:** Phil R. Bell, Brian J. Pickles, Caleb M. Brown.

**Resources:** Michael Rampe, Caleb M. Brown.

**Software:** Michael Rampe.

**Supervision:** Phil R. Bell, Brian J. Pickles.

**Validation:** Phil R. Bell, Brian J. Pickles, Paul Durkin, Caleb M. Brown.

**Visualization:** Phil R. Bell, Brian J. Pickles, Sally Hurst, Michael Rampe, Paul Durkin, Caleb M. Brown.

**Writing – original draft:** Phil R. Bell, Brian J. Pickles, Caleb M. Brown.

**Writing – review & editing:** Phil R. Bell, Brian J. Pickles, Sarah C. Ashby, Issy E. Walker, Sally Hurst, Michael Rampe, Paul Durkin, Caleb M. Brown.

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
