## [Decision Letter · Decision Letter 0]

25 Mar 2025

Dear Dr. Pickles,

Thank you for submitting your manuscript to PLOS ONE. After careful consideration, we feel that it has merit but does not fully meet PLOS ONE’s publication criteria as it currently stands. Therefore, we invite you to submit a revised version of the manuscript that addresses the points raised during the review process.

We look forward to receiving your revised manuscript.

Kind regards,

Ulrich Joger

Academic Editor

PLOS ONE

Journal Requirements:

2. For studies involving third-party data, we encourage authors to share any data specific to their analyses that they can legally distribute. PLOS recognizes, however, that authors may be using third-party data they do not have the rights to share. When third-party data cannot be publicly shared, authors must provide all information necessary for interested researchers to apply to gain access to the data. (https://journals.plos.org/plosone/s/data-availability#loc-acceptable-data-access-restrictions)

4) All necessary contact information others would need to apply to gain access to the data.

4. We note that Figures 1 and 2 in your submission contain [map/satellite] images which may be copyrighted. All PLOS content is published under the Creative Commons Attribution License (CC BY 4.0), which means that the manuscript, images, and Supporting Information files will be freely available online, and any third party is permitted to access, download, copy, distribute, and use these materials in any way, even commercially, with proper attribution. For these reasons, we cannot publish previously copyrighted maps or satellite images created using proprietary data, such as Google software (Google Maps, Street View, and Earth). For more information, see our copyright guidelines: http://journals.plos.org/plosone/s/licenses-and-copyright.

1. You may seek permission from the original copyright holder of Figures 1 and 2 to publish the content specifically under the CC BY 4.0 license. 

Additional Editor Comments:

Your manuscript is nearly perfect. Minor corrections are listed by reviewer 1 in a commented copy of your manuscript. Please follow his advice. I refrain from asking a second reviewer as there is nothing to add.

Reviewers' comments:

Reviewer's Responses to Questions

**Comments to the Author**

1. Is the manuscript technically sound, and do the data support the conclusions?

Reviewer #1: Yes

2. Has the statistical analysis been performed appropriately and rigorously?

Reviewer #1: Yes

3. Have the authors made all data underlying the findings in their manuscript fully available?

Reviewer #1: Yes

4. Is the manuscript presented in an intelligible fashion and written in standard English?

Reviewer #1: Yes

Reviewer #1: The manuscript is about dinosaur tracks from Ceratopsians, Ankylosaurians, and Theropods from Dinosaur Provincial Park in Alberta, Canada, which were discovered and excavated in summer 2024. Interestingly researches concerning dinosaur tracks from this area are rather sparse, what makes this publication a worthy contribution to dinosaur ichnotaxa research. The manuscript is very legible written and only a very few comments were made (see attached PDF file).

**Do you want your identity to be public for this peer review?** For information about this choice, including consent withdrawal, please see our Privacy Policy

Reviewer #1: No

---

## [Author Response · Author response to Decision Letter 1]

11 Apr 2025

On behalf of my co-authors I would like to say that we are extremely grateful for the positive reviews from the Editor and Reviewer 1 and have endeavoured to make all of the suggested changes. We also note our responses to journal requirements. Please see specific responses to Reviewer 1 and Journal requirements below.

---

Response to Reviewer 1

Page 5, lines 88-90: This chapter is too short. Which were the methods you used, for example to measure the tracks?

Page 6, line 115: This should be shifted to the methods chapter

Page 6, line 125: Should also be shifted to the methods chapter

Page 6, line 135: methods

Page 7, line 141: methods

Page 7, line 157: methods chapter

Response: The intention here is for the Materials and Methods section to be divided up into subsections. The font sizes appear to be correct as per PLoS ONE requirements (Level 1: Materials and Methods = 18; Level 2 subheadings = 16) but we agree that this 2pt font size change is not the most obvious change between sections. As Excavation, Mapping, etc. are all subheadings under Materials and Methods we have made this clearer by adding “methods” to each of those subheadings.

Page 12, line 264: alternative verb?

Response: Thank you for catching this repetition, we have changed the wording to avoid this and it now reads “Three distinct types of invertebrate traces were present, all preserved as medium-grained sandstone infill with a host matrix.”

Page 21, lines 479-480: Are they probably a new ichnospecies?

Response: We appreciate that the morphology of the tyrannosaurid tracks discussed here could be interpreted as representative of a novel ichnotaxon. However, as demonstrated by Enriquez et al. (2021), there is substantial ontogenetic variation in tyrannosaurid tracks that has yet to be accounted for in the Dinosaur Park Formation. Until a better understanding of such changes is available (based on new discoveries - which, given the current rate of discovery, may not be in the too-distant future!) we prefer a conservative stance and leave the DPP tyrannosaur tracks in open nomenclature.

Page 22, Lines 484-486: Was it documented before via photogrammmetry?

Response: Yes, the trace was observed during the excavation and Fig. 10 is the photograph associated with the trace when it was found. We have now noted in the text that it was documented before photogrammetry.

---

Response to Journal requirements

Response: Having thoroughly checked the documentation we believe our article meets all specified style requirements as per PLoS ONE instructions. The main issue appears to have been with Figure file naming which has now been corrected. If there are any that we have missed, please note exactly which files are causing problems and I will fix them right away.

2. For studies involving third-party data, we encourage authors to share any data specific to their analyses that they can legally distribute. PLOS recognizes, however, that authors may be using third-party data they do not have the rights to share. When third-party data cannot be publicly shared, authors must provide all information necessary for interested researchers to apply to gain access to the data. (https://journals.plos.org/plosone/s/data-availability#loc-acceptable-data-access-restrictions)

4) All necessary contact information others would need to apply to gain access to the data.

Response: Due to the sensitive nature of the site, which may be vulnerable to vandalism and is set to be further excavated in July 2025 we have removed the site coordinates from the Geological Setting section. We hope that the following data availability statement is acceptable but please let us know if you have any requested changes.

Data Availability Statement: All data necessary to replicate the analyses are presented in the manuscript and its Supporting Information files. The Skyline Tracksite is located within the Dinosaur Provincial Park Palaeontological Preserve. Due to the sensitivity of the site coordinates cannot be publicly shared but were provided to reviewers. Interested researchers may apply to gain access to the full site coordinate data, which are on file at the Royal Tyrrell Museum of Palaeontology, Drumheller, Alberta. Site number L2467. Please contact RTMP Collections: https://tyrrellmuseum.com/research/collections

Response: Thank you for catching this. We have now removed the ethics part from the Acknowledgements section and moved it into the start of the Materials and Methods section.

4. We note that Figures 1 and 2 in your submission contain [map/satellite] images which may be copyrighted. All PLOS content is published under the Creative Commons Attribution License (CC BY 4.0), which means that the manuscript, images, and Supporting Information files will be freely available online, and any third party is permitted to access, download, copy, distribute, and use these materials in any way, even commercially, with proper attribution. For these reasons, we cannot publish previously copyrighted maps or satellite images created using proprietary data, such as Google software (Google Maps, Street View, and Earth). For more information, see our copyright guidelines: http://journals.plos.org/plosone/s/licenses-and-copyright.

Response: Figure 1 is made up of photographs taken by one of the lead authors (PRB) so we do not believe that copyright needs to be requested, but we have included a form noting that PRB is happy for the photos to be used as Fig 1 in the manuscript. Figure 2 is a set of site maps made in Adobe Illustrator (V 23.0.6) by study co-author CMB using the hand drawn maps that were created during the field work. Again we do not believe that there are any copyright issues here as no proprietary data was used, but we have included a form noting that CMB made these maps to be used as Fig. 2 in the manuscript.

Response: We have gone through all of the references and corrected the doi’s to put them in the correct format, which appeared to be the problem here. Please see the tracked changes version of the manuscript for all changes to reference formatting. As far as we are aware none of the cited papers have been retracted, but if you have noted any please could you indicate which ones?

---

## [Editor Report · Decision Letter 1]

4 May 2025

A ceratopsid-dominated tracksite from the Dinosaur Park Formation (Campanian) at Dinosaur Provincial Park, Alberta, Canada

PONE-D-25-02384R1

Dear Dr. Pickles,

We’re pleased to inform you that your manuscript has been judged scientifically suitable for publication and will be formally accepted for publication once it meets all outstanding technical requirements.

Kind regards,

Ulrich Joger

Academic Editor

PLOS ONE
---

## [Editor Report · Acceptance letter]

PONE-D-25-02384R1

PLOS ONE

Dear Dr. Pickles,

I'm pleased to inform you that your manuscript has been deemed suitable for publication in PLOS ONE. Congratulations! Your manuscript is now being handed over to our production team.

Kind regards,

on behalf of

Dr. Ulrich Joger

Academic Editor

PLOS ONE